# RVM-GSM: Classification of OCT Images of Genitourinary Syndrome of Menopause Based on Integrated Model of Local–Global Information Pattern

**DOI:** 10.3390/bioengineering10040450

**Published:** 2023-04-06

**Authors:** Kaiwen Song, Haoran Wang, Xinyu Guo, Mingyang Sun, Yanbin Shao, Songfeng Xue, Hongwei Zhang, Tianyu Zhang

**Affiliations:** The Key Laboratory of Geophysical Exploration Equipment, Ministry of Education, College of Instrumentation and Electrical Engineering, Jilin University, Changchun 130012, China

**Keywords:** deep learning, genitourinary syndrome of menopause (GSM), image classification, optical coherence tomography (OCT), convolutional neural network (CNN), vision transformer (ViT)

## Abstract

Genitourinary syndrome of menopause (GSM) is a group of syndromes, including atrophy of the reproductive tract and urinary tract, and sexual dysfunction, caused by decreased levels of hormones, such as estrogen, in women during the transition to, or late stage of, menopause. GSM symptoms can gradually become severe with age and menopausal time, seriously affecting the safety, and physical and mental health, of patients. Optical coherence tomography (OCT) systems can obtain images similar to “optical slices” in a non-destructive manner. This paper presents a neural network, called RVM-GSM, to implement automatic classification tasks for different types of GSM-OCT images. The RVM-GSM module uses a convolutional neural network (CNN) and a vision transformer (ViT) to capture local and global features of the GSM-OCT images, respectively, and, then, fuses the two features in a multi-layer perception module to classify the image. In accordance with the practical needs of clinical work, lightweight post-processing is added to the final surface of the RVM-GSM module to compress the module. Experimental results showed that the accuracy rate of RVM-GSM in the GSM-OCT image classification task was 98.2%. This result is better than those of the CNN and Vit models, demonstrating the promise and potential of the application of RVM-GSM in the physical health and hygiene fields for women.

## 1. Introduction

Genitourinary syndrome of menopause (GSM) is a term used to indicate the symptoms and signs of reproductive tract atrophy, urinary tract atrophy, and sexual dysfunction in women during the menopausal transition or in the post-menopausal period, due to reduced levels of estrogen and other hormones [1]. The focal symptoms include dryness and itching of the vulva and vagina, urinary tract infections, and pain or difficulty in sexual intercourse. These conditions can have serious negative impacts on the quality of life of the woman concerned, especially the quality of her sexual life. Treatment for GSM is based on the severity of the patient’s symptoms. Vaginal lubricants and vaginal moisturizers are generally recommended for mild–moderate patients, and topical vaginal estrogen therapy is generally recommended for moderate–severe patients. Some of the latter patients may not be able to use estrogen-containing medications, for medical reasons, and may be recommended laser treatment, such as microablative $CO^{2}$ laser and non-ablative erbium laser treatments. In a report on more than 900 routinely screened women, 84% of the women were screened for symptoms included in GSM for the six years after menopause [2]. Contemporary society urgently needs a method that can distinguish GSM symptoms well.

OCT is a new optical imaging technology, based on the principle of low coherence interference, which is, in turn, primarily based on the use of the Michelson Interferometer. OCT combines confocal scanning technology and low-coherence interferometry technology, and uses image processing technology to achieve non-contact imaging. It works similarly to B-type ultrasound imaging [3]. It is a non-contact, non-invasive, imaging technique that perfectly combines high resolution and large imaging depth to achieve a biopsy-like role in histopathology [4]. Currently, OCT technology has been widely used in clinical diagnosis in medical fields, such as ophthalmology [5], dentistry [6] and dermatology [7], because of the advantages described above, but the use of this technology in fields such as GSM and other gynecological diseases is still rare. Therefore, it is necessary to develop the application of OCT technology in fields such as GSM.

Generally, the diagnosis of GSM disease mainly depends on the doctor’s understanding of the disease and clinical experience. The inspection process takes a long time, the judgment of the result is full of subjectivity, and improving the accuracy of the inspection is not possible. The advent of automated computer-aided diagnostic systems (CADs) has reduced this problem to some extent. CAD has the advantages of accurate quantitative analysis of radiological images and repeatable processing, reducing the workload of imaging physicians and improving the efficiency of hospital visits. Since the information contained in OCT images is more complex than that in CT images, it is more difficult to recognize OCT images, so it is necessary to develop CAD technology to assist in the diagnosis and treatment of GSM. Early CAD technology was based on classical image processing and related mathematical theory analysis techniques. Later, the rapid development of neural networks further improved the integration of CAD and machine learning. However, due to the fact that the features in machine learning are manually designed, this can lead to processing results that do not achieve the expected goals [8]. Therefore, deep learning technology, with automatic learning of target feature information, has become a better solution. At this stage, with the rapid development of deep learning technology, many achievements have been made in the following fields: computer vision [9,10], object detection [11], natural language processing (NLP) [12], and speech recognition [13,14]. In the field of image classification, especially, the effects are better than those of classical machine learning [15].

Convolutional neural network (CNN) is a widely used and varied deep learning network. It has made many breakthroughs in the field of computer vision [16], especially in image segmentation, detection, and classification tasks, and has achieved good results [17,18,19]. Recently, the transformer model was applied in the field of NLP and achieved great success. After being processed by Google, a Vision Transformer(ViT) model was launched that uses a pure transformer to complete computer vision tasks. On the premise of using large amounts of data for pre-training, it is superior to the latest convolutional networks. At the same time, when migrating the pre-training model to medium or small dataset tasks, it can also perform better than CNN [20].

In this study, we introduce RVM-GSM using RegNet, ViT-P, and Multilayer Perceptron (MLP). We apply it to a GSM OCT image dataset, collected using OCT equipment, to propose a model that can distinguish between GSM disease screening stages. In RVM-GSM, a parallel network of RegNet and ViT-P is used. The RegNet module is used to extract local features of GSM images, and the ViT-P module is used to extract global variables of GSM images, followed by feature fusion and classification operations. The whole training process is illustrated in Figure 1.

(a) GSM-OCT images were obtained through our own equipment and used as training data. Section 3.2 explains the details of the dataset.

(b) In the data training phase, the training dataset was first preprocessed, i.e., enhanced and standardized. The preprocessed data were input into the RVM-GSM module, where the RegNet and ViT-P modules were used to extract local and global features, respectively. Then, the extracted features were input into the MLP module for fusion and classification. In addition, we introduced lightweight postprocessing to compress the model.

(c) The test images were provided to RVM-GSM for classification.

(d) The performance of the RVM-GSM module was evaluated by calculating the accuracy, recall, sensitivity and specificity values.

This article’s GSM computer-aided diagnostic task, based on deep learning, contributes the following:

(1) A new neural network framework (RVM-GSM) is proposed, which mainly uses CNN and Transformer modules for GSM lesion feature fusion to improve overall classification performance, while using lightweight postprocessing to compress the model and reduce complexity.

(2) Our RVM-GSM achieved a good result of 98.2% in the GSM lesion classification task, outperforming traditional CNN and ViT network models.

(3) The RVM-GSM network not only has good detection ability in GSM-OCT images, but also has good performance in retinal disease classification, verifying the generalization ability of the network.

The article is organized as follows. Section 2 provides an introduction to the related work involved in the task. Section 3 elaborates data augmentation methods, datasets, model frameworks, experimental settings, and evaluation methods. Section 4 presents the experimental results of deep learning on the GSM data, a comparison with the advanced CNN model, and extended experiments. Finally, Section 5 summarizes the work in this article and provides future directions for exploration.

## 2. Related Work

With the improvement of data volume and computing power, artificial intelligence (AI) has become more and more closely connected with various industries. Therefore, a large number of researchers have applied AI to the diagnosis and efficacy prediction of clinical diseases. OCT is an important factor in medical disease diagnosis and imaging technology, so many researchers have combined AI with OCT technology as an essential method that assists doctors in diagnosis and treatment. The initial combination was the application of machine learning in AI to OCT retinal disease monitoring for disease screening. In 2014, Srinivasan et al. [21] proposed a classification algorithm for disease screening in OCT retinal images. The algorithm combines the sparsity and distribution curvature of the retina to remove speckle noise carried by OCT images, and uses gradient direction multiscale histograms as feature vectors in support vector machines (SVM) for the training. The accuracy of the classifier for images acquired from patients with aged macular degeneration (AMD), patients with diabetic macular edema (DME), and normal subjects reached 100%, 100%, and 86.67%, respectively. In 2016, Wang et al. [22] used the public dataset provided by Srinivasan et al. to classify and analyze OCT images, and used the linear collocation pattern (LCP) feature and sequence minimum optimization (SMO) algorithm of OCT images for recognition and classification. Using a multi-scale method for feature extraction enables LCP features to be calculated on multiple scales, achieving complementarity between local and overall information extracted from the OCT images. However, for OCT image classification tasks, the following two shortcomings are relevant: (1) the noise associated with image datasets varies greatly, for objective reasons; (2) feature labeling relies too much on the proficiency of ophthalmologists in this field. These shortcomings make the above methods less feasible in large datasets. Machine learning algorithms do not have good operability and fault tolerance for complex data, which means that, in practical applications, these algorithms may not achieve satisfactory results in the face of complex OCT images.

Therefore, researchers have focused on deep learning algorithms that can perform autonomous learning. In 2018, Kermany et al. [23] applied the pretrained convolution neural network (CNN), GoogleLeNet_V3, using the method of transfer learning, to the classification tasks of the following four types of OCT images: choroidal neovascularization (CNV), DME, vitreous warts (DRUSEN), and normal. At the same time, they tested chest X-ray images in an expanded experiment, and the accuracy rates of the two experiments reached 96.6% and 92.8%, respectively. In 2019, Feng et al. [24] also used the method of transfer learning to fine tune the visual geometry group 16 (VGG-16) neural network to accurately classify AMD and DME in OCT images. By adjusting the network structure, the demand for the number of OCT images in the dataset was reduced and the differences between different data sets were enhanced. Due to the many achievements achieved in OCT images using the method of transfer learning, more teams began to design networks based on the features carried by OCT images. In 2019, Wang et al. [25] focused on the impact of noise interference in OCT images on classification results. First, they used the FsatNlMeans method and the bilateral filter method to preprocess the images, and then used the more advanced model, CliqueNet, for pretraining. The accuracy rates in testing the two data sets reached 99% and 98.6%, respectively. In 2019, Li et al. [26] introduced a model fusion method to improve the generalization ability of models in OCT classification tasks. They replaced the standard convolution in the original ResNet50 architecture with inflated convolution, and used different, improved, ResNet50 architectures for integration operations. After calculating the average value of the probabilities of different network outputs, they obtained experimental results with an accuracy of 97.9%. The model is robust and effective, but the tenfold cross-validation method used in experiments increases the computational load of the machine. In 2021, Potapenko et al. [27] proposed a model based on a convolutional neural network (CNN) to address issues such as the need for extensive preparation and the long time taken in image labeling before training OCT data. Images in their datasets did not require preprocessing and labeling. Considering that the training labels came directly from clinical datasets without relabeling or extensive preprocessing, the experiments demonstrated the potential capabilities of CNNs in extracting imaging data, providing more possibilities for the future development of deep learning algorithms related to OCT images. In 2023, Karthik et al. [28] increased the contrast of residual connections based on ResNet architecture to generate cleaner feature maps for the classifying of tasks. Applying this to OCT datasets involving eight types of disease, the classification accuracy improved 1.6% on the original ResNet framework. However, most OCT images have characteristics such as sparse features in the transform domain, wide distribution range, and regular distribution. Therefore, when extracting features from CNN networks, insufficient attention to global information can lead to feature loss. Therefore, in 2021, Wang et al. [29] first introduced Visual Transformer (ViT) into the field of medical OCT images, and, based on this, proposed a ViT-P model for GSM lesion screening. This model replaces the 16 × 16 convolution in the original data preprocessing section with an ultra-thin model based on the ViT model, improving the diversity of the network. In sample processing, combining an empty convolutional network with adversarial neural networks to generate samples for experiments avoids some problems. The accuracy of screening for GSM and UCSD datasets reached 99.9% and 99.69%, respectively.

As can be seen from the reference text, most research work uses CNN to extract local features of images and then the classification tasks are performed. However, GSM-OCT images have rich global information, and this method of extracting features through convolution can lead to excessive attention to local features in the network model, resulting in incomplete restored data. Therefore, the ViT model was introduced to compensate for the shortcomings of CNN’s attention mechanism to help the model extract global information from GSM-OCT images, combining local and global informational features for more accurate analysis. The network thus built can simultaneously utilize the advantages of CNN and ViT to accurately screen GSM-OCT for diseases.

## 3. Material and Methods

### 3.1. Data Augmentation

The GSM-OCT dataset contains inconsistent samples of normal, GSM and under treatment (UT) data, and such category imbalance means the data fail to meet the network training requirements. Therefore, there is a need for a method to generate a large number of OCT images on the existing basis to ensure that the network can learn enough pathological knowledge. For this reason, the deep convolutional confrontation generation network (B-DCGAN), with a large receptive field, proposed in our previous article, was adopted [29]. The advantage of B-DCGAN over traditional DCGAN is that the traditional convolutional kernel in the discriminator is replaced by extended convolution, which further expands the receptive field and achieves the collection of more pathological information in the image without changing the resolution. By using B-DCGAN, we solved the problem of uneven sample sizes for various classes in the original dataset.

### 3.2. Datasets

The light source of the OCT system, used to collect data in the laboratory, uses a vertical cavity surface emitting laser (VCSEL) scanning source (Thorlabs Inc., Newton, NJ, USA, 100 kHz, 100 nm bandwidth) with a central wavelength of 1.3 μm. The detection part is a self-developed proximal scanning flexible endoscope OCT imaging probe with a diameter of 1.2 mm. The sample arm of the OCT system consists of an optical fiber swivel and imaging probe, which are used to transmit light to the tissue to be tested and to receive the information returned. The reference arm consists of a fiber optic collimator, a lens group, and a mirror that sets a corresponding delay distance.

The original experimental data contained three types, namely, normal, GSM, and UT. Experimental data were collected from 69 normal subjects, 46 patients, and 15 subjects undergoing treatment. The subjects in these three states provided 2600, 1526 and 720 OCT images, respectively, totaling 4846 images. Then, using the data enhancement method described in Section 3.1, the B-DCGAN data enhancement model was used to process the three types of data to meet the data volume required for the final experiment. The three types of experimental data volumes were: normal (3451 sheets), GSM (2699 sheets), and UT (2320 sheets). The labels used in the experimental data were all from labels made by doctors in the relevant fields, based on their years of medical experience. The test set used in the experiment consisted of 200 OCT images from 5 selected subjects in each subject category, and the remaining images were used as training sets.The details of the data are shown in Figure 2.

(1) GSM: OCT images of menopausal (55–70 years of age) patients with urogenital syndrome without any treatment, with a VET range of 88.4 to 111 μm and a mean VET of 104.5 μm.

(2) UT: OCT images of postmenopausal female patients with urogenital syndrome who had undergone fractional pixel CO${}_{2}$ laser treatment (repeat scans of treated postmenopausal female patients 4–6 weeks after laser treatment, as well as baseline OCT scans). Results of clinical trials have shown that VET generally ranges from 134 μm to 174.4 μm, with a mean VET of 154 μm.

(3) Normal: OCT images without GSM. Subjects were non-menopausal females with a mean age of 33 years, with normal menstruation at the time of the past year, reporting no changes in the menstrual cycle, and for whom the physician gave a VET range of 263.1–345 μm, consistent with the normal label. The mean VET was 318 μm.

### 3.3. RegNet

There have been many classic architectures in the development of CNN, which can be divided into two directions: Handcraft and Neural Architecture Search (NAS). In terms of Handcraft, from the early LaNet and AlexNet to the recent well-known VGG, Inception, ResNet, MobileNet, etc., the proposal of each network includes the views of the author of the network architecture. Network search aims to directly find the best network architecture through neural architecture search (NAS) [30], such as NaeNet. The traditional NAS method, based on individual estimation (sampling one network at each evaluation), has the following drawbacks: (1) lack of flexibility; (2) poor generalization; (3) poor interpretability.

Therefore, we focused on the RegNet network structure proposed by Radosavovic et al. [31]. In the lightweight network field, the RegNet model, with low floating point operations (FLOPs), can also achieve good results, with performance comparable to MobileNetV2 and ShuffleNetV2. Compared with EfficientNet, which belongs to the classification network, the experimental results using RegNet have a lower error rate, and the computing speed on the GPU is nearly five times higher. Compared with NasNet, the main advantage is that the overall strategy of network search is progressive. It improves the quality of the space, while designing in a relatively primitive and unconstrained space, and looks for simpler or stronger models. Therefore, we used RegNet to extract local features of lesions on GSM-OCT images. GSM-OCT images have the characteristics of complex information, strong feature contrast, and small feature space. Therefore, the use of RegNet is very suitable for GSM-OCT images. Figure 3a shows the structure of the RegNet module. It is mainly composed of three parts, namely, a convolutional layer, four stage stacks, and a classifier. In the convolution layer, the convolution kernel was 3×3, stride = 2, and the convolution kernel size was 32. In the next section, at each stage, as shown in Figure 3b, the height and width of the input feature matrix are reduced by half. Each stage is composed of a series of block stacks. The first block of each stage has a group convolution with a step of 2 (on the main branch, as in Figure 3c) and a normal convolution (on the shortcut branch, as in Figure 3d), while the remaining blocks have a convolutional step of 1, similar to ResNet. The final classifier consists of a global average pooling layer and a fully connected layer.

### 3.4. ViT-P

ViT is the first model in the field of computer vision that uses transformer encoders instead of standard convolution [20,32]. The ViT model for image classification can be divided into two phases: the feature extraction phase and the classification phase. In the feature extraction phase, in order to process a two-dimensional image into a one-dimensional sequence, the original image $x\in R^{H\times W\times C}$ is reconstructed to obtain a two-dimensional patch sequence $x_{p}\in R^{N\times (p^{2}\times C)}$. C is the number of channels of the image, (H×W) is the size of each image patch, and $N=HW/p^{2}$ is the sum of the number of patches with the same length as the input sequence of the transformer encoder. The transformer uses an invariant hidden vector D to traverse all layers, and all patches are spread to the D dimension, the patch embeddings ofwhich are mapped by a trainable linear projection. Then, in order to preserve the location information, one-dimensional locations are combined with a sequence of input vectors and the patch embedding transformer encoder is selected as input. The structure of the transformer encoder is shown in Figure 4a and is composed of several alternative Multi-Headed Self Attention mechanism (MHSA) blocks [33] and a Multilayer Perceptron (MLP) [34]. Layernorm (LN) is used in front of each layer, connected to the following block by a residual connection. The MLP module consists of a nonlinear Gaussian error linear unit (GELU) activation function connecting the two network layers. Finally, in the classification phase, the output features of the feature extraction phase are passed through a fully connected layer composed of MLPs to obtain the classification confidence.

ViT-P structure has two advantages over ViT structure. On the one hand, changes have been made to the original ViT preprocessing section, which uses a special two-dimensional convolutional operation to achieve image segmentation. Its complex and large step convolution kernel design violates CNN’s design philosophy. Therefore, ViT-P made changes on this basis and proposed a slim model to replace it. As shown in Figure 4b, the complex large convolutions in each step are replaced with multiple small convolutions, while using smaller convolution cores. This reduces the complexity of the network and improves GSM image feature reuse without too much change in model parameters and complexity. On the other hand, ViT-P adds a channel attention mechanism to the original ViT structure, using SE modules to help the network learn important features in each channel, and assigns different weight coefficients to distinguish the importance between features, making up for the disadvantage that ViT cannot obtain the importance of image channels. Therefore, we used the ViT-P module in GSM-OCT images to capture the global information of the image. Figure 4a shows the structure of the ViT-P module. Using patch embedding to linearly transform the preprocessed image data, the output results were normalized and processed by multi-head attention, summation, LN and MLP modules, and this was repeated L times. The structure of patch embedding is shown in Figure 4b. The design is of a slim module composed of four stage modules and a 1 × 1 convolutional layer. This design improves the diversity of the network and reduces complexity. The details of the stage used are shown in Figure 4c. In order to reduce the complexity of the network, only 3×3 convolutions are used in the network backbone, while the more concise 1×1 convolution is used in the branch network. This design improves the reuse of GSM features. At the end of the stage structure, a channel attention mechanism is introduced to enhance important features and suppress other features. The most classic channel attention mechanism used here was the SE module. The SE module is shown in Figure 4d. The model first converts the input characteristic graph into a vector with a size of C×1×1 through global average pooling, which represents the global distribution of responses on the channel. Two fully connected layer structures are used to build correlation between channels. The input feature is dimensioned down to 1/16 of the input dimension through the first fully connected layer, and then the input feature dimension is restored to the original dimension through the second fully connected layer. The sigmoid function is used for channel by channel feature selection.

### 3.5. RVM-GSM

For GSM-OCT images, we propose a deep learning framework called RVM-GSM, which forms a new network architecture by adding CNN and ViT in parallel. CNN has the ability to locally induce bias, which can efficiently model local features and achieve good results on small sample datasets. Therefore, the introduction of CNN brings local inductive bias to the overall network, solving multiple problems, such as the inability of the ViT model to perform well on small sample datasets and the difficulty of initializing training that requires a large amount of data to compensate. The detailed process of the RVM-GSM network framework is shown in Figure 5.

In the preprocessing stage, the B-DCGAN network is first used for data enhancement to address the imbalance in the numbers of the three types of images. Then, each image is processed using a random horizontal flipping function, having a 50% probability of being flipped horizontally, and increasing the generalization ability of the RVM-GSM algorithm.

Then, the preprocessed dataset is input into the RVM-GSM module, and local and global features are extracted through the RegNet and ViT-P modules, respectively. Local features of lesions are extracted from GSM-OCT images in the RegNet module. In designing the RegNet network model the respective advantages of Handcraft and NAS were combined, increasing the ability to generalize the model. The network thus designed can better complete the extraction of lesion features and aim at identifying the characteristics of the GSM-OCT images, such as sparse lesion features, wide distribution range, and regular distribution. In the ViT-P module, first, the Slim model is used to reduce network complexity and to improve the reusability of GSM-OCT image focus features. Then, the SE module is combined to obtain the importance of each channel of information in the network, achieving the acquisition of GSM-OCT global information.

The 196 image lesion features captured by the ViT-P module and the 2048 image lesion features captured by the RegNet module are then input to the MLP module for feature fusion and classification. The final output layer can predict the classification results based on the calculation. In the MLP module, the GELU activation function is used to introduce nonlinear factors to the neurons, so that the neural network can use more nonlinear models to enhance the expression ability of the neural network and to achieve better classification capabilities. We chose the front-end fusion strategy [35] to directly concactenate the 196 and 2048 features input to the MLP module into 2244 features. The fused generated lesion feature information is then used to classify the lesion images. Finally, according to the actual needs of clinical work, we carried out lightweight postprocessing to compress the model [36]. We used a lightweight module to reduce the number of bits needed to change each weight value from 32 bits to 24 bits. Despite reducing the model parameters and model complexity, the classification performance was basically unchanged.

### 3.6. Experimental Settings

The image size of the generator in B-DCGAN was set to 64 × 64, the training samples input to the discriminator were resized to the same size, and the output was 0 or 1 (0 meant the discriminator was recognized as synthetic data and 1 meant the discriminator was recognized as a real training sample). The initial learning rate was 0.0002 and the images generated after 50 independent training sessions of the network were used as supplementary data. The Adam optimizer was also used to update and compute the model network parameters to approach, or to reach, the optimal values, thus minimizing the loss function.

To train RVM-GSM, we initialized the learning rate as 0.001 and the BatchSize as 32. Other parameters used were the original parameters in ImageNet-21K for weighted training, only changing the number of FC nodes in the last layer, and performing 50 training iterations. Patch Embedding was only a slim model part for initial training. To keep the training fair, the rest of the nets used the above parameter settings.

### 3.7. Statistical Analysis Method

In order to avoid adverse effects, such as misclassification of subjects due to factors such as low screening accuracy, four experimental evaluation indices, namely accuracy (ACC), precision, sensitivity and specificity values, were used as an important basis for reference. ACC, Precision, Sensitivity and Specificity values were defined as follows: (1)$$Accuracy=\frac{TP+TN}{TP+TN+FP+FN}$$
(2)$$Precision=\frac{TP}{TP+FP}$$
(3)$$Sensitivity=\frac{TP}{TP+FN}$$
(4)$$Specificity=\frac{TN}{TN+FP}$$

True positive (*TP*): the number of positive samples classified as positive by the model; false negative (*FN*): the number of positive samples classified as negative by the model; false positive (*FP*): the number of negative samples misclassified as positive by the model; true negative (*TN*): the number of negative samples classified as negative by the model. Overall sensitivity (OS): the weighted average of the sensitivities. Overall precision (OP): the weighted average of each precision.

### 3.8. Experimental Environment

The experimental code was implemented in python 3.9 under the Python framework, and all experiments were conducted under the Windows 10 operating system on a machine with CPU Intel Core i7-10875H 2.30 GHz, GPU NVIDIA RTX 3070, 16 GB of RAM, and both NVIDIA Cuda v11.1 and cuDNN v11.1 acceleration libraries.

## 4. Experiments and Results

Based on our evaluation method, we compared RVM-GSM with RegNet, ViT, and ResNet50 on the GSM-OCT image dataset for classification task processing. Based on the results, their performances in classification were evaluated and discussed.

### 4.1. Results on GSM Dataset

The ResNet [37] is designed with reference to the VGG19 network, and modified based on it. A residual unit was added through the short circuit mechanism, and a short circuit connection was added between every two layers of the ordinary network to form residual learning. In the experiments, a 50-layer ResNet was constructed, containing a convolutional layer, a pooling layer and a fully connected layer. Thee RegNet network [31] combines the advantages of manual design and NAS, and the main difference between this network and traditional methods is that, instead of designing individual network instances individually, the design space of the parameterized network population is designed. In the experiments, RegNet networks consisting of convolutional, pooling, and fully connected layers were built to screen GSM-OCT image datasets. The experimental design of ViT networks was based on our previous work [29].

Table 1 shows the classification results obtained by applying different neural networks to the GSM dataset. The results show that different neural network models were effective for screening GSM lesions, and could accurately identify GSM, UT, and Normal images. RVM-GSM had the highest accuracy, reaching 98.2%, while RstNet50 had the worst classification results, achieving 93.5% accuracy. At the same time, the effectiveness of the independently designed CNN model (ResNet) was a bit worse than the network search model based on the NAS mechanism (RegNet).

The confusion matrix corresponding to the four models is given in Figure 6. As shown in Figure 6a, the ResNet50 model had poor diagnostic results for two categories, GSM and UT, while the Normal category could be effectively identified. As shown in Figure 6b, the RegNet model effectively identified the two categories UT and Normal, while the identification results for the GSM category had many errors. As shown in Figure 6c, the ViT model was better in recognizing the GSM class compared to ResNet50 and RegNet, but was worse in recognizing the UT and Normal classes compared to the previous two models. As shown in Figure 6d, the RVM-GSM model effectively identified GSM and Normal classes and checked a little better than the first three network models did, but was slightly less effective in identifying UT classes. ResNet50 and RegNet were less effective in diagnosing GSM classes, mainly because of the use of pooling layers in the model structure, which led to the loss of much valuable information and ignored correlation between the whole and the part. That is, the sense field was small, which allowed local features to be extracted well, but the network still had some defects when global features needed to be extracted. Our proposed RVM-GSM has the advantage of combining the strong local feature extraction ability of CNN and the strong global feature extraction ability of ViT, so it achieved better results in all three categories of images.

### 4.2. Extended Experiments

We further tested the performance of the RVM-GSM model on the public Srinivasan OCT dataset and the OCT Retinal Disease Open Dataset (UCSD dataset), and demonstrated that the effectiveness of the RVM-GSM network architecture is not only valid on private GSM-OCT datasets. The experimental setup for the extended experiment was based on GSM images, and then fine-tuned according to actual data requirements.

#### 4.2.1. Experiments on the Extension of the Srinivasan 2014 Dataset

The Srinivasan 2014 dataset was publicly provided by Duke University and contains images of three labeled retinal diseases, namely, Normal, age-related macular degeneration (AMD), and diabetic macular degeneration (DME). These three OCT images have different image features for classifier learning and disease screening, as shown in Figure 7. This extended experiment proved that the RVM-GSM model not only performed well in GSM classification tasks, but was also effective in OCT ophthalmic dataset classification tasks.

In the OCT image classification task, the images were divided into a training dataset and a validation dataset. The distribution of the number of image types is shown in Table 2. The training dataset contained 302 normal images, 618 AMD images and 996 DME images. The validation dataset contained 105 images for each category.

According to Table 3, the parameters of RVM-GSM in the diagnosis of retinal diseases were superior to VGG-16 and ResNet-v1, with the resulting ACC reaching 98.3%, which was about 12.18% and 3.6% higher than those for VGG-16 and ResNet-v1, respectively. Lee et al. [38] used a modified VGG-16 in the last layer using KNN. As shown in Table 3, RVM-GSM achieved more advanced results, seen by evaluating the metrics Accuracy, Sensitivity and Specificity. In addition, we trained ResNet50-v1 [37] to classify the Srinivasan 2014 dataset, and the results are shown in Table 3, wherein it can be seen that it was slightly less effective compared to RVM-GSM. The RVM-GSM model focuses on both local and global information, which allow it to effectively classify different classes of images with a large degree of similarity, as shown in Table 3. The RVM-GSM model effectively classified the three retinal OCT images of NORMAL, DME and AMD.

#### 4.2.2. Experiments on the Extension of the UCSD Dataset

The UCSD dataset contains four tagged retinal disease images, Normal, CNV, DME and DRUSEN, as shown in Figure 8. These samples were produced by the Healthy Eye Institute at the University of California, San Diego, the California Retina Research Foundation, the Center for Ophthalmic Medicine, the Shanghai First People’s Hospital, and the Tong Ren Eye Center in Beijing. The allocation of data usage is shown in Table 4. The number of normal images was 26,320, the number of CNV images was 37,215, the number of DME images was 11,358, and the number of DRUSEN images was 8606. Due to the selection of each sample through data preprocessing, the number of samples used for training was different from the number of samples provided in the public data set, which reduced the number of samples with high noise in various types. At the same time, the test set contained 242 OCT images of four categories, totaling 968. The sample size in the test set was 512 × 496, while the training set was full of variables.

To compare the effectiveness of the models, we investigated the effect of using the RVM-GSM model with other more advanced CNN models (e.g., AlexNet [39] and LACCN [35]) on the UCSD dataset. First, we set the weight attenuation factor λ to 0.0002 to ensure the accuracy and effectiveness of the experimental results, BatchSize was set to 32, and then we referred to the LACNN network in [35] to set the attenuation factor of the first two layers of FC to 70% in the experiment. We changed the training epoch to 20, and other parameter settings did not change. The training results are shown in Table 5.

As shown in Table 5, RVM-GSM outperformed AlexNet and LACCN in terms of parameters for the UCSD dataset, and the ACC of its diagnostic results showed an improvement of about 26.12% and 9.32% over AlexNet and LACCN, respectively. Extensive experiments on retinal OCT data sets show that RVM-GSM can be very effectively applied to not only GSM classification tasks, but also to other types of disease classification tasks. At the same time, when the sample base is expanded, the accuracy of the CNN model decreases to a certain extent, due to its limitations in focusing on local information, while the RVM-GSM model does not have similar problems, because it focuses on both local and global information. Therefore, the RVM-GSM model can be applied to diseases with more characteristic information, which is helpful in practical applications.

## 5. Conclusions

This paper proposes a deep learning-based framework called RVM-GSM for GSM disease classification tasks. RVM-GSM uses the RegNet module and ViT-P module to extract local and global features of lesions, respectively, and uses MLP to fuse the two features for classification tasks. It combines the advantages of a transformer and a CNN to fully utilize the advantages of both neural networks in the diagnosis process. Finally, according to clinical needs, a lightweight post-processing compression was performed on the model, with the model parameter size compressed from 120 MB to 80 MB. In the two types of experiments of GSM disease diagnosis and retinal disease screening, the overall accuracy of RVM-GSM reached 98.2% and 98.3%, respectively, surpassing the SOTA model.

In our future work, we will deeply explore the generalization capability of the RVM-GSM model from the perspective of the model and from the perspective of data. We will also try to deploy the RVM-GSM model into a real clinical environment to explore the application value, or potential, of this network and to promote the clinical deployment of neural network models.

## Figures and Tables

**Figure 1 bioengineering-10-00450-f001:**
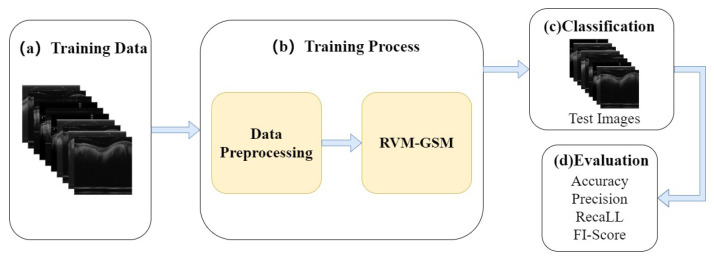
Overall structure of the experimental model. (**a**) Training Data. (**b**) Training process. (**c**) Classification. (**d**) Evaluation.

**Figure 2 bioengineering-10-00450-f002:**
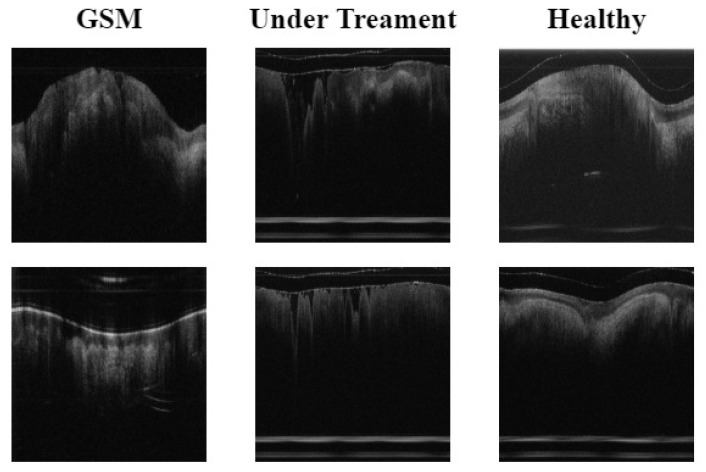
OCT images of Genitourinary syndrome of menopause (GSM), Under Treatment (UT), and Healthy people.

**Figure 3 bioengineering-10-00450-f003:**
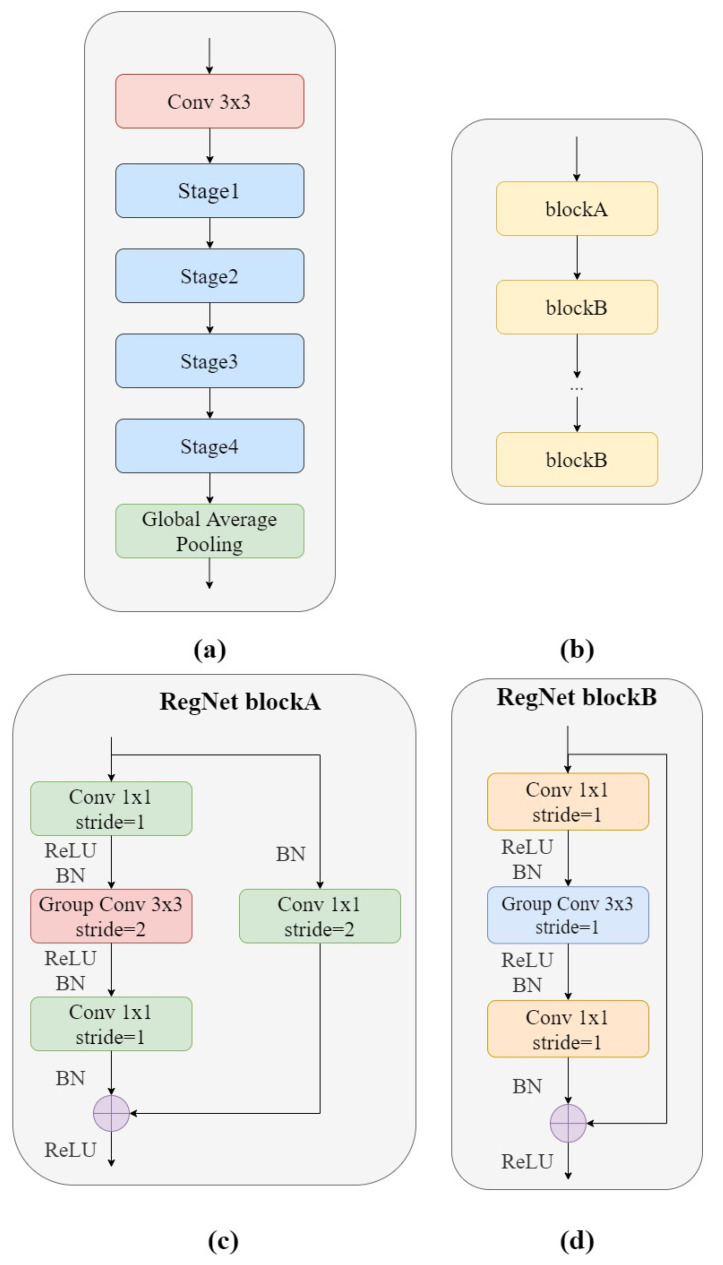
(**a**) RegNet structural framework; (**b**) Detailed structure of stage; (**c**) RegNet block A, Group Conv stride = 2; (**d**) RegNet block B, Group Conv stride = 1.

**Figure 4 bioengineering-10-00450-f004:**
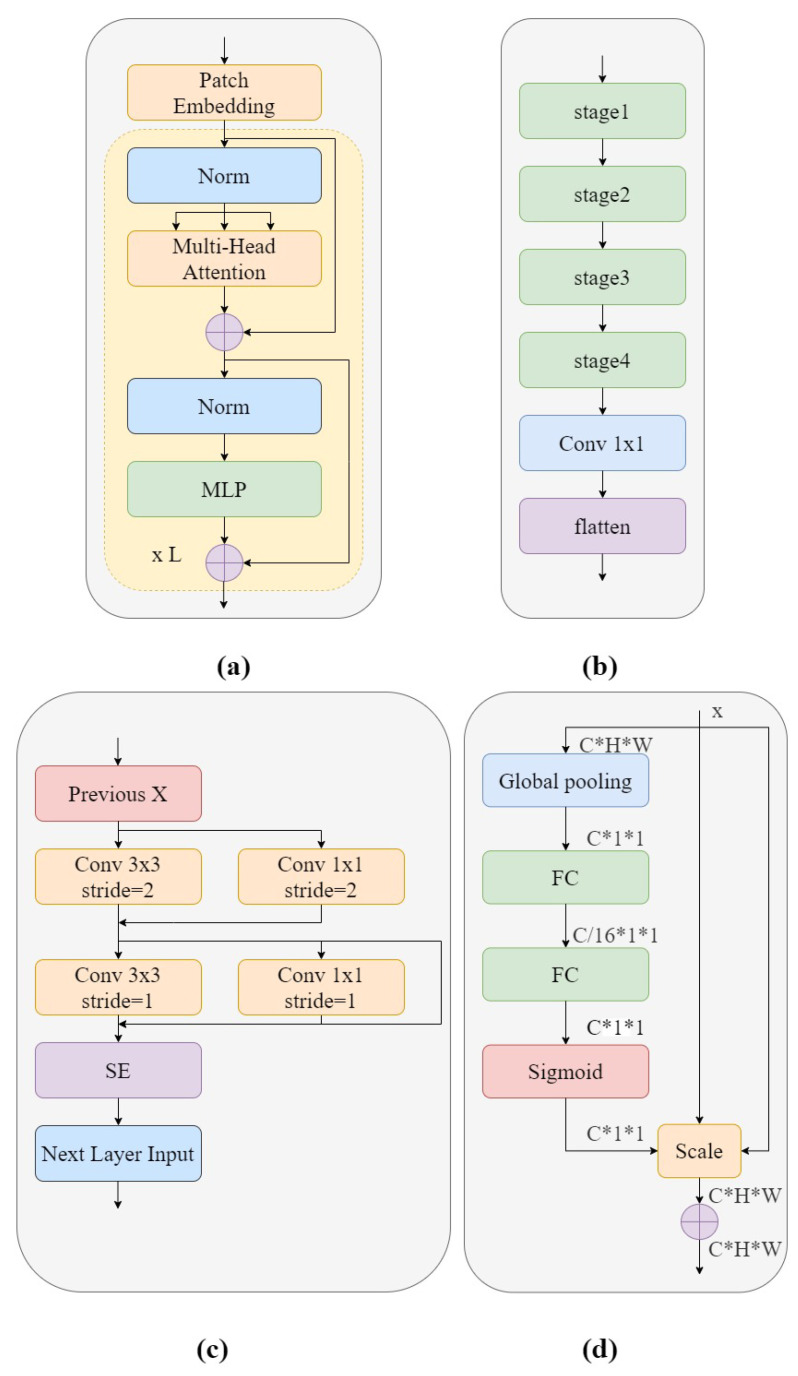
(**a**) The structure of the transformer encoder; (**b**) ViT-P patch embedding; (**c**) Detailed structure of each ViT-P patch embedding stage; (**d**) SE model.

**Figure 5 bioengineering-10-00450-f005:**
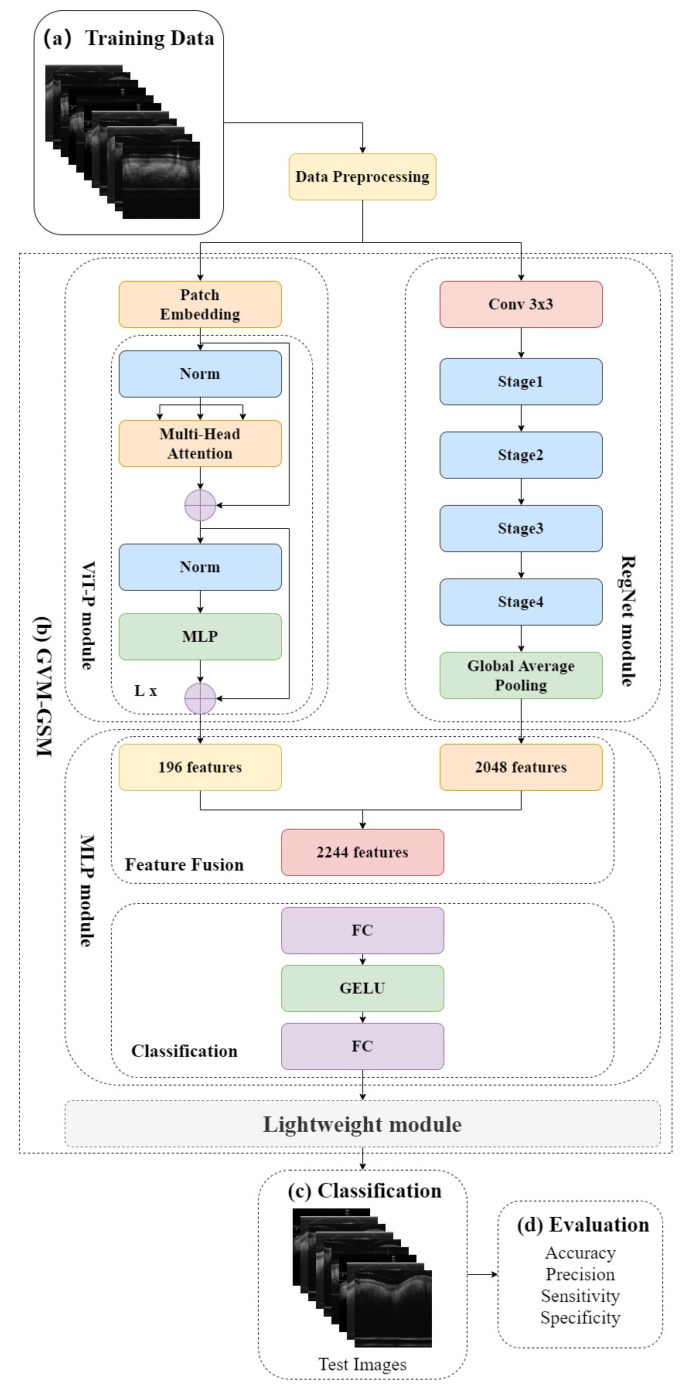
The structure of RVM-GSM. (**a**) shows the GSM-OCT image used to train the RVM-GSM; (**b**) shows the main body of RVM-GSM, which consists of three parts. First, RVM-GSM is designed to perform global and local feature extraction using ViT-P and RegNet modules. Then, feature fusion and classification are performed by the MLP module. In total, 196 features were captured by the ViT-P module and 2048 features were finally captured by the RegNet module. The 196 and 2048 features extracted by the ViT-P module and RegNet module are first fused in the MLP module to generate 2244 features. Then, MLP is applied for classification. Finally, the model is postprocessed as follows: lightweight module is used to achieve compression; (**c**) RVM-GSM is used to classify the test set images; (**d**) the performance of the RVM-GSM check is evaluated in terms of accuracy (ACC), precision, sensitivity and specificity values.

**Figure 6 bioengineering-10-00450-f006:**
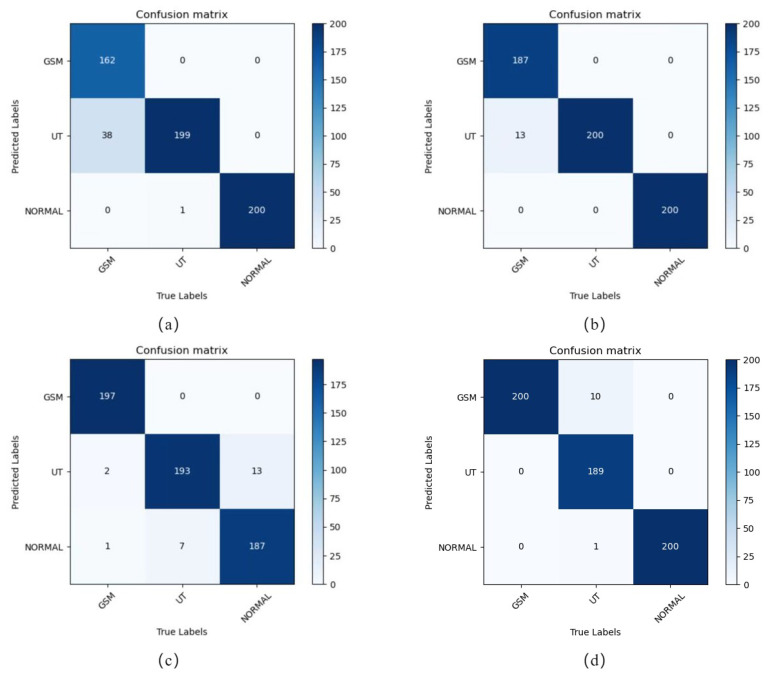
Neural network model confusion matrix. (**a**) ResNet50, (**b**) RegNet, (**c**) ViT, and (**d**) RVM-GSM.

**Figure 7 bioengineering-10-00450-f007:**
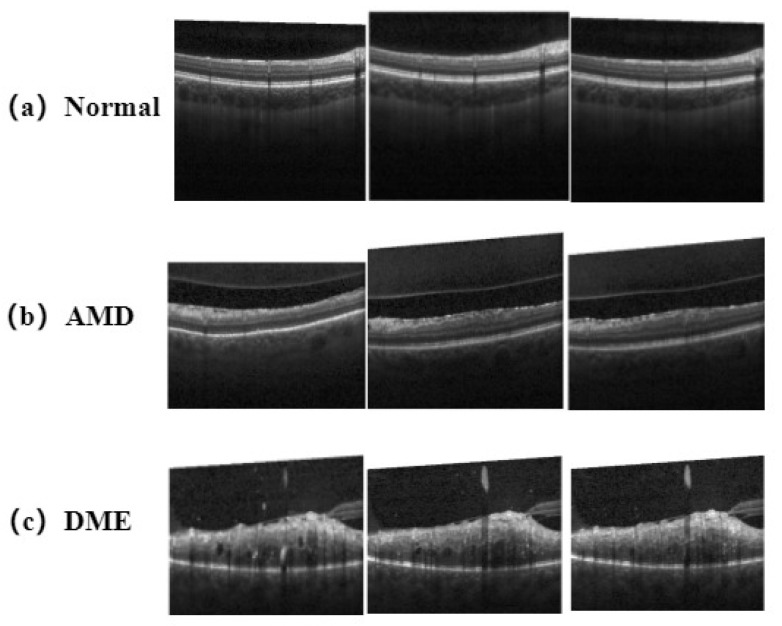
Sample category visualization for the Srinivasan 2014 dataset. (**a**) Normal: The normal retina presents regular depressions; (**b**) Age-related Macular Degeneration (AMD): Due to hemorrhage and fluid exudation caused by macular degeneration, a new layer of membrane forms on the retina, and edema forms at the lesion site. New layers and irregular protrusions appear; (**c**) Diabetic Macular Degeneration (DME): Due to the accumulation of retinal fluid, retinal thickness increases and reflectivity decreases, resulting in an increase in central thickness and a decrease in intensity of the image.

**Figure 8 bioengineering-10-00450-f008:**
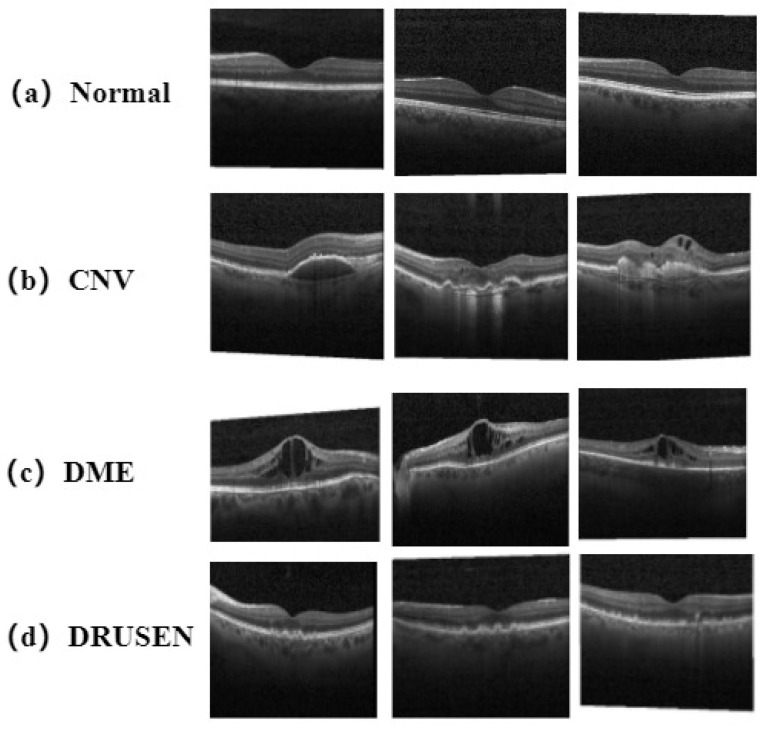
Sample category visualization for the UCSD dataset. (**a**) Normal: The normal retina presents regular depressions; (**b**) Choroidal Neovascularization (CNV): Due to thickening and enhanced reflection of the epithelial layer/choroidal capillaries in the retina, protrusions appear in the image in a fusiform or elliptical shape; (**c**) DME: Due to the accumulation of retinal fluid, retinal thickness increased and reflectivity decreased, resulting in an increase in central thickness and a decrease in intensity of the image; (**d**) Vitreous Warts (DRUSEN): In the corresponding area of the glass membrane wart, a low reflection area caused by shielding can be seen, presenting a special image of a hat-like high reflection boundary accompanied by a low reflection area of a lower void-like cavity.

**Table 1 bioengineering-10-00450-t001:** Comparison results of models on the GSM dataset.

Method	Classes	Precision	Sensitivity	Specificity	ACC
ResNet50	GSM	1.00	0.81	1.00	0.935
	UT	0.84	0.995	0.905	
	Normal	0.995	1.00	0.998	
RegNet	GSM	1.00	0.935	1.00	0.978
	UT	0.939	1.00	0.968	
	Normal	1.00	1.00	1.00	
ViT	GSM	1.00	0.985	1.00	0.962
	UT	0.928	0.965	0.962	
	Normal	0.959	0.935	0.98	
RVM-GSM	GSM	0.971	1.00	0.98	0.982
	UT	0.99	0.992	0.971	
	Normal	0.992	0.966	0.999	

**Table 2 bioengineering-10-00450-t002:** Data settings of the Srinivasan 2014 dataset.

Image Type	Training	Test	Sum
NORMAL	302	105	407
AMD	618	105	723
DME	996	105	1111

**Table 3 bioengineering-10-00450-t003:** Comparison results of models on the Srinivasan 2014 dataset.

Method	Precision	Sensitivity	Specificity	Accuracy
VGG-16 [38]	86.31	84.63	91.54	87.63
ResNet-v1 [37]	92.15	94.92	97.46	94.92
RVM-GSM	96.81	95.52	99.07	98.3

**Table 4 bioengineering-10-00450-t004:** Data settings of the UCSD dataset.

Image Type	Training	Test	Sum
NORMAL	26,320	242	226,562
CNV	37,215	242	37,457
DME	11,358	242	11,600
DRUSEN	8606	242	8848

**Table 5 bioengineering-10-00450-t005:** Experimental comparison of the effects of models on the ucsd dataset (%).

Method	Classes	Precision	Sensitivity	Specificitvity	ACC
AlexNet [39]	Drusen	51.5 ± 5.2	58.5 ± 7.3	93.9 ± 2.9	78.1 ± 0.7
	CNV	94.0 ± 2.2	84.5 ± 4.3	95.3 ± 1.4	
	DME	75.9 ± 4.7	69.8 ± 4.8	95.7 ± 1.7	
	Normal	86.2 ± 3.3	91.5 ± 7.8	90.7 ± 3.4	
LACCN [35]	Drusen	70.0 ± 5.7	72.5 ± 7.9	95.9 ± 2.1	90.1 ± 1.4
	CNV	93.5 ± 1.3	89.8 ± 4.5	95.1 ± 1.6	
	DME	86.4 ± 1.6	87.5 ± 1.5	98.0 ± 0.3	
	Normal	94.8 ± 1.1	97.3 ± 1.0	97.4 ± 0.5	
RVM-GSM	Drusen	95.0 ± 0.5	98.0 ± 0.3	97.3 ± 1.2	98.5 ± 0.23
	CNV	98.3 ± 0.6	98.1 ± 0.2	98.7 ± 0.4	
	DME	99.1 ± 0.4	97.9 ± 0.7	98.4 ± 0.3	
	Normal	98.2 ± 0.5	98.6 ± 0.31	97.6 ± 1.2	

## Data Availability

Not applicable.

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
