# Peer review of "RVM-GSM: Classification of OCT Images of Genitourinary Syndrome of Menopause Based on Integrated Model of Local–Global Information Pattern"

_bioengineering, 2023, doi:10.3390/bioengineering10040450_

Round 1
Reviewer 1 Report
Although this manuscript attempts to describe deep learning for the diagnosis of menopausal urogenital syndrome from OCT images, it excessively includes deep learning retinal images.
For example, it is suitable that "2. Related Work" will be included in "1. Introduction". After including "2. Related Work" in "1. Introduction", it is necessary to organize "1. Introduction".
Also, "4.2. Extended experiments" also does not fit the contents of this manuscript.
Therefore, this manuscript is unsuitable for publication as it requires extensive revision.
Reviewer 2 Report
Comments are in the attachment.

Reviewer 3 Report
The paper has introduced a deep learning based RVM-GSM network network for the classification task of GSM-OCT images. There is no way to understand what RVM stands for till 3rd page. And many abbreviations in the manuscript are short of interpretations. It is not a smart way of writing articles. Please check very carefully throughout the paper to make sure every abbreviation has been clearly explained when they show up for the first time.
The manuscript has used public resource to show that RVM-GSM can be integrated in OCT devices to perform automatic imaging classification work and the results in the table 1 showed the accuracy is 98.2%. For this result, how many sample images have been used for the training and test? In page 8, it is hard to understand how authors have successfully used RVM-GSM for using ViT with small sample datasets without problem. And, how small for the sample datasets?
Except above major concerns, all figures shown in the manuscripts are too small and the words in the figure is impossible to read.
Round 2
Reviewer 1 Report
In this manuscript, the reviewer's comments were well reflected. So, this manuscript is suitable for publication.
Reviewer 2 Report
Authors modified the manuscript according to the comments.
I have no more comments.
Reviewer 3 Report
The author has done great work on the revision.
The revised manuscript has improved figure panels with high resolution. And the revised figure legends offered much cleared details. All results supported the conclusion well. My most comments have been well addressed.